# What is the impact of structural changes in society on diabetes self-management and trajectories of HbA1c? A cohort study before, during and after the COVID-19 pandemic in people with diabetes treated at outpatient clinics

**Martin Gillies Banke Rasmussen** [1,2]*, **Emilie Just-Østergaard**[1], **Jacob Volmer Stidsen**[1], **Ingrid Willaing**[3,4], **Grete Skøtt Pedersen**[1]

**1** Steno Diabetes Center Odense, Odense University Hospital, Odense, Denmark, **2** Research Unit for Exercise Epidemiology, Department of Sports Science and Clinical Biomechanics, Center of Research in Childhood Health, University of Southern Denmark, Odense, Denmark, **3** Steno Diabetes Center Copenhagen, Herlev, Denmark, **4** Department of Health Services Research, Institute of Public Health, University of Copenhagen, Copenhagen, Denmark

\* martin.gillies.banke.rasmussen@rsyd.dk

## Abstract

### Background

The impact of COVID-19-related changes in diabetes self-management and trajectories of HbA1c throughout COVID-19 is not fully understood. Here, we describe HbA1c trajectories, changes in diabetes self-management and their association before, during and after the COVID-19 pandemic (2019–2022).

### Methods

During the spring of 2021, we invited 13,641 outpatients from diabetes clinics in the Region of Southern Denmark to complete a questionnaire regarding changes in diabetes self-management during COVID-19. We linked the questionnaire and registry HbA1c data from before, during and after the COVID-19 pandemic and conducted multivariable adjusted linear mixed-effect regression to assess the association between changes in diabetes self-management and HbA1c.

### Results

5,581 (40.9%) people responded to the questionnaire (median age: 65 years, males: 59.7%). HbA1c decreased in people with type 2-diabetes and was unchanged for people with type 1-diabetes (interaction: p < 0.001). The majority of people reported unchanged diet (65–71%) and usual medication taking (89–90%). No changes in physical activity were reported by 43%, while 42% reported decreased physical activity. HbA1c trajectories did not differ according to change in physical activity and

**Data availability statement:** The data contains highly sensitive information, which cannot be de-identified. Researchers may obtain a license and ethics approval such that data can be made available. Researchers are required to have their project registered on the internal directory at the Region of Southern Denmark (https://regionsyddanmark.dk/fagfolk/forskning/patientjournaloplysninger-til-forskning-statistik-eller-planlaegning) and apply for data retrieval at the regional business intelligence department (https://regionsyddanmark.dk/om-region-syddanmark/organisation/omrader-og-stabe/afdelinger-og-omrader-i-den-centrale-administration/dokumentation-og-ledelsesinformation).

**Funding:** The author(s) received no specific funding for this work.

**Competing interests:** The authors have declared that no competing interests exist.

change in diet intake, while taking medication more regularly was associated with a decrease in HbA1c, from approximately 65/66 mmol/mol (8.1/8.2%) to 60/61 mmol/mol (7.6/7.7%) in both diabetes types.

## Conclusions

During COVID-19, HbA1c trajectories differed between diabetes types. Most of the sample maintained usual diabetes self-management, although some decreased physical activity levels. Improved medication taking was associated with decreased HbA1c. This information is crucial for health professionals, in order to provide support aimed at reducing HbA1c.

## Introduction

The rapid spread of coronavirus SARV-CoV-2 (COVID-19) was declared a pandemic by the World Health Organization on the 11th of March 2020 [1], which resulted in immediate nationwide lockdowns in many countries, with risk of social isolation and disruption of everyday routines. In Denmark, as part of the lockdown, social gatherings were limited to no more than ten people. Thus, sports clubs, restaurants and shopping malls were closed. Most restrictions were lifted during the summer of 2020, but a second lockdown was implemented later the same year, which continued into March of 2021 [2]. Moreover, face-to-face hospital visits were limited and replaced with telehealth contacts.

During the early pandemic, people with diabetes were assessed to be at higher risk of contracting a COVID-19 infection and also at higher risk of severe morbidity and mortality compared to people without chronic disease [3]. Moreover, evidence from the initial period of the pandemic showed a marked decrease in HbA1c testing in people with diabetes in the UK and the United States [4,5] and in the number of diabetes outpatient visits [5].

Diet, physical activity and medication taking, i.e., treatment-related factors with known impact on glycemic levels, were reported to be impacted by the pandemic in people with diabetes. A review including studies of people with diabetes suggested both positive and negative changes in diet during the early pandemic [6], while many studies suggest decreased physical activity [7–15]. In a Danish sample of almost 1.400 people with either Type 1- (T1) or Type-2-diabetes (T2D) 40.1% reported exercising less than usual [9], and during a one-year follow-up 50.6% reported less physical activity, relative to the time before COVID-19 [16]. However, the level of physical activity appeared to be unchanged in most of Australian [17] and Indian [18] samples of people with T2D. A study of Malaysian patients with T2D suggested improved medication taking of glucose-lowering drugs (based on the number of reimbursed prescriptions) and improved glycemic levels during the pandemic [19]. Overall, these studies suggest non-uniformity in the behavioral response to the pandemic in people with diabetes.

The COVID-19 pandemic potentially had a negative impact on glycemic levels in people with diabetes [20,21], although such an association has not been found

consistently [20,22]. This impact has likely been heterogeneous and may, to some extent, be explained by differences in changes in diabetes self-management, but this remains to be explored in more detail. Some studies have found that changes in the level of physical activity and diet during COVID-19 have been associated with increases in Hba1c levels during COVID-19 [7,12,23], while others have reported no association [10]. The abovementioned studies, however, have not analyzed HbA1c trajectories throughout the pandemic according to changes in diet, physical activity and medication taking.

Therefore, the aims of the study were 1) to describe HbA1c trajectories before, during and after the COVID-19 pandemic (2019–2022), 2) to describe changes to diet, physical activity and medication taking during the first year of the COVID-19 pandemic and 3) to associate these to HbA1c trajectories before, during and after the COVID-19 pandemic (2019–2022) in adults treated at Danish diabetes outpatient clinics. We hypothesized that negative changes in diabetes self-management would be associated with increased glycemic levels.

## Methods

### Population

We invited people (≥16 years) with a registered diabetes diagnosis (DE10*-DE14*) and at least one outpatient contact between March 2020 and March 2021 at all clinical departments treating people with diabetes, at one of the five public hospitals in the Region of Southern Denmark (OUH Odense, OUH Svendborg, Sydvestjysk Sygehus, Sygehus Lillebælt and Sygehus Sønderjylland) to complete a survey.

### Study design and data collection

We conducted a retrospective cohort study based on a survey completed during the spring of 2021, i.e., approximately one year into the COVID-19 pandemic, and laboratory data (between the 11th of March 2019 and the 30th of April 2022). The survey was distributed electronically via personal digital mail (e-Boks), which is available to all citizens in Denmark, on the 31st of March 2021–13,641 patients. Postal letters were sent to those exempt from digital mail (15% of the invited participants). In a cover letter to the survey, we informed participants that survey completion implied consent to participation in the survey and further consent to data-usage for research. Initial non-response triggered an online reminder and an additional second reminder on the 23rd of April for the age group <50 years, who had the lowest response rates in preliminary summaries of responses. Civil registration numbers for distribution of the questionnaire were obtained from the regional database of patient records (EPJ). Later, we linked HbA1c data from the regional laboratory database to survey data, via the civil registration number.

We obtained approval from the Region of Southern Denmark, in accordance with the Data Protection Regulation and the Data Protection Act (project-id: 22/57532). The Medical Directors of all Regional Hospitals approved use of data from medical records (i.e., HbA1c) on behalf of the patients, according to Danish law. We had access to the data from the 1st of January 2023. The data was stored on a safe server in the Region of Southern Denmark that complies with instructions of the General Data Protection Regulation. Data was pseudoanonymized before statistical analyses were conducted.

### Exposures

Questions regarding diet, physical activity and medication taking during the pandemic were translated into Danish and slightly modified from an American questionnaire, with permission from Fisher et. al. (2020) [8]. We also included a question regarding changes in bodyweight (S1 Appendix). The respondents were asked to rate changes in diabetes self-management relative to pre-pandemic conditions. Each question was rated on a 7-point Likert scale, ranging from a large change in either direction. E.g. for diet, the categories ranged from; "Eating much more", "eating somewhat more", "eating slightly more", "eating approximately the same amount", "eating slightly less", "eating somewhat less" and "eating much

less". Due to data limitations, the seven categories were aggregated to three categories: any change in a negative direction, no change and any change in a positive direction.

## Outcome

We obtained HbA1c assessments registered between 11th of March 2019 (a year prior to national lockdown) and the 30th of April 2022 (after the pandemic) from the regional laboratory database. The database includes all HbA1c assessments (laboratory codes: NPU27300) analyzed at large clinical laboratories, collected for any purpose at visits to General Practitioners, inpatient and outpatient clinics in the Region of Southern Denmark.

## Covariates and background variables

The survey included questions on diabetes type (T1D/T2D/Other (not specified in subtypes)), year of diabetes onset, and highest educational attainment (six categories). Age and sex were obtained from the regional database of patient records. In the survey we also asked respondents to report whether they had diabetes-related complications: retinopathy, neuropathy, nephropathy, foot ulcers and cardiovascular disease. We imputed "no complication" in case of missing. Number of complications was categorized as; 0, 1, 2 and ≥3 complications. The respondents also reported status, relating to COVID-19 infection and vaccination, and if they had been sent home from work due to COVID-19. Lastly, we collected data on diabetes distress using the Problems Areas in Diabetes (PAID)-5 scale, and well-being using the World Health Organization (WHO)-5 scale. The PAID-5 scale ranges from 0–20, where a score of ≥8 indicates possible diabetes-related emotional distress [24]. The WHO-5 scale ranges from 1–100, where 0 is the lowest level of well-being and 100 the highest level of well-being [25].

## Statistical analysis

We described the sample characteristics by frequencies, percentages, medians, and lower and upper quantiles. To assess the construct validity of change in diet and physical activity we cross-tabulated each variable with the reported change in bodyweight (S2 Appendix).

We used linear mixed-effects regression with patient-specific random intercepts to assess the association between one-year change in diet, physical activity and medication taking and HbA1c. We computed cubic splines with knots at percentiles (5 27.5 50 72.5 95) of the dates of HbA1c assessment. Models were restricted to those with ≥2 HbA1c measurements, and stratified by diabetes type. However, due to data sparsity and because the findings were materially the same, the results are presented unstratified. Stratified results are presented in S2 Appendix. We imputed missing data (in 10 datasets) for exposure variables and covariates using Multiple Imputation by Chained Equations [26] with age, sex, diabetes type and number of diabetes-related complications as predictors (S2 Appendix). Data was assumed missing at random. Because there were not enough people with other diabetes types, only descriptive statistics and crude analysis were conducted for this group (S2 Appendix).

Firstly, for each exposure variable, we conducted analyses only adjusting for age (years) and sex (male/female) (Crude analyses, S2 Appendix). Secondly, we additionally adjusted for diabetes duration (years), number of diabetes-related complications (0, 1, 2, ≥3) and educational attainment (elementary, High School or vocational, short or bachelor level, master's degree or higher, other/not declared). To test differences in HbA1c trajectories between exposure groups, we included an interaction term between each spline term and the exposure of interest, and tested all of these in a Wald test. Thirdly, we conducted three sensitivity analyses for each of the main analyses. Firstly, we restricted our sample to those with ≥1 HbA1c measurement each year (2019−2022). In the second sensitivity analysis, we restricted our sample to those with ≥1 HbA1c measurements between the 11th of March 2020 and the 31st of July 2020 (the early pandemic), who might be a highly selected group (a group who could not postpone hospital treatment during lockdown). In the third sensitivity analysis, we specified the knots at five pivotal dates during COVID-19; the 11th of March 2020 (first national lockdown), the

1st of December 2020 (just before children were sent home from school and around the time of the first vaccinations), the 1st of April 2021 (around when the Delta Variant became prevalent), the 10th of September 2021 (when COVID-19 was no longer deemed a disease posing societal health risk) and the 1st of April 2022 (post COVID-19).

We deemed α ≤ 0.05 statistically significant. The analyses were conducted in Stata version 17.0 SE.

## Results

### Flow

Altogether 13,641 people were invited to participate in the survey and approximately four out of 10 (40.9%) responded (Fig 1). People with T2D constituted most of the analytic sample, with 351 more people with T2D compared to T1D.

Overall, the respondents differed in several respects according to diabetes type (Table 1). Those with T2D were oldest, the highest proportion of males, had the largest proportion with elementary school as highest educational level, and constituted the group with the largest percent with at least one diabetes-related complication. Those with T1D had the longest

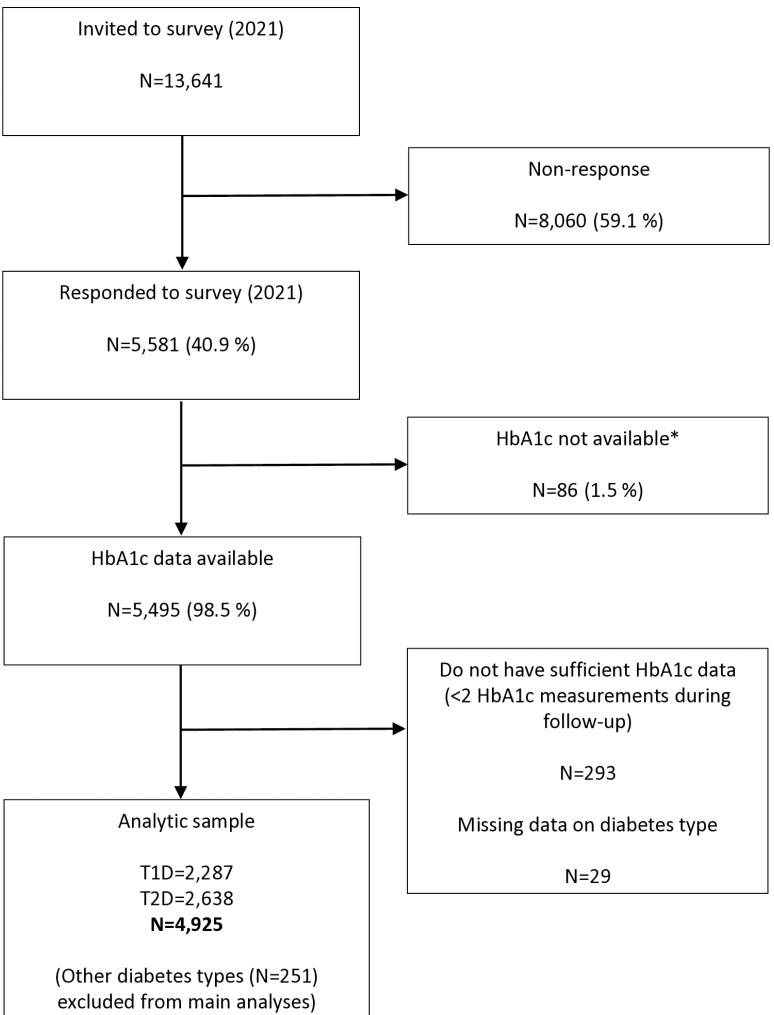

**Fig 1. Flow chart from invitation to analyses.** *For administrative reasons HbA1c was not available for a small subset of the population.

**Table 1. Characteristics of questionnaire respondents with type 1-diabetes and type 2-diabetes (N = 4,925).**

| | T1D | T2D | Total |
|---|---|---|---|
| **N** | 2,287 | 2,638 | 4,925 |
| **Age (years), median (q1-q3)** | 59 (48-69) | 69 (60-75) | 65 (54-73) |
| **Men (n (%))** | 1,226 (54%) | 1,735 (66%) | 2,961 (60%) |
| **Cohabitation status** | | | |
| Partner or married | 1,507 (71%) | 1,582 (68%) | 3,089 (69%) |
| Single | 552 (26%) | 705 (30%) | 1,257 (28%) |
| Other/not declared | 71 (3.3%) | 48 (2.1%) | 119 (2.7%) |
| Missing | 157 | 303 | 460 |
| **Educational attainment** | | | |
| Elementary | 310 (14%) | 555 (21%) | 865 (18%) |
| High school or vocational | 964 (43%) | 1,032 (40%) | 1,996 (41%) |
| Short or bachelor level | 610 (27%) | 504 (19%) | 1,114 (23%) |
| Masters degree or higher | 200 (8.8%) | 136 (5.2%) | 336 (6.9%) |
| Other/not declared | 184 (8.1%) | 372 (14%) | 556 (11%) |
| Missing | 19 | 39 | 60 |
| **Diabetes duration (years), median (q1-q3)** | 26 (16-39) | 17 (10 –24 ) | 21 (11-31) |
| Missing | 19 | 39 | 243 |
| **Number of complications** | | | |
| 0 | 1,543 (68%) | 1,411 (54%) | 2,954 (60%) |
| 1 | 456 (20%) | 742 (28%) | 1,198 (24%) |
| 2 | 195 (8.5%) | 331 (12.5%) | 526 (11%) |
| ≥3 | 93 (4.1%) | 154 (5.8%) | 247 (5%) |
| **Has been infected with the coronavirus** | | | |
| No/do not know | 2,180 (98%) | 2,493 (97%) | 4,673 (97%) |
| Yes | 44 (2.0%) | 84 (3.3%) | 128 (2.7%) |
| N missing | 63 | 61 | 124 |
| **Vaccinated against the coronavirus** | | | |
| No | 1,254 (57%) | 1,066 (42%) | 2,320 (48%) |
| Yes | 966 (44%) | 1,504 (59%) | 2,470 (52%) |
| N missing | 67 | 68 | 135 |
| **Sent home due to the coronavirus** | | | |
| No/not relevant | 1,596 (72%) | 2,281 (89%) | 3,877 (81%) |
| Yes | 616 (28%) | 274 (11%) | 890 (19%) |
| Months sent home due to the coronavirus (months), median (q1-q3) | 4 (2 –7 ) | 3.5 (2 –6 ) | 4 (2 –7 ) |
| N missing | 75 | 83 | 158 |
| **PAID-5 (score 0–20), median (q1-q3)** | 5 (2 –9 ) | 5 (1 –8 ) | 5 (2 –9 ) |
| N missing | 137 | 170 | 307 |
| **WHO-5 (score 0–100), median (q1-q3)** | 64 (44-80) | 64 (44-80) | 64 (44-80) |
| N missing | 138 | 194 | 332 |

*The descriptive statistics above are of the observed data and thus the number of observed observations for each variable vary. The number of missing values are presented for context. Descriptive statistics for the diabetes category "Other" are not included here, but can be found in S2 Appendix.*

diabetes duration. Regarding COVID-19, those with T2D had the largest percentage of COVID vaccination and had the lowest proportion of people sent home from work due the coronavirus.

The median (lower quantile-upper quantile) number of HbA1c assessments was 8 [6–10] and 10 [7–12] for T1D and T2D, respectively.

### HbA1c trajectories

We found a statistically significant interaction between HbA1c trajectories and diabetes type (p < 0.001). We observed a tendency towards a decrease in mean HbA1c in people with T2D, while for people with T1D, mean HbA1c, appeared mostly unchanged. Additionally, we noted a trend towards a decrease in HbA1c from the summer of 2020 to the summer of 2021 among those with other diabetes types, although mean HbA1c appeared to increase during the remaining follow-up time (Fig 2).

### Changes to diet, physical activity and medication

Numerically, most respondents reported no change in diet, physical activity or medication taking compared to pre-COVID-19 (Fig 3). Furthermore, approximately as many reported eating less or eating more, and equivalent proportions reported decreased and unchanged physical activity (both between 41.3 and 43.7%). Few reported taking medication less regularly, and just below one in ten reported taking their medication more regularly. To assess the degree of construct validity of the variables change in diet and change in physical activity we created two contingency tables in which each variable was tabulated against changes in bodyweight. There appeared to be an association in the expected directions for both variables; a decrease, no change and an increase in bodyweight was associated with eating less and being more physically activity, no change in diet and physical activity and eating more and being less physically active, respectively (S2 Appendix).

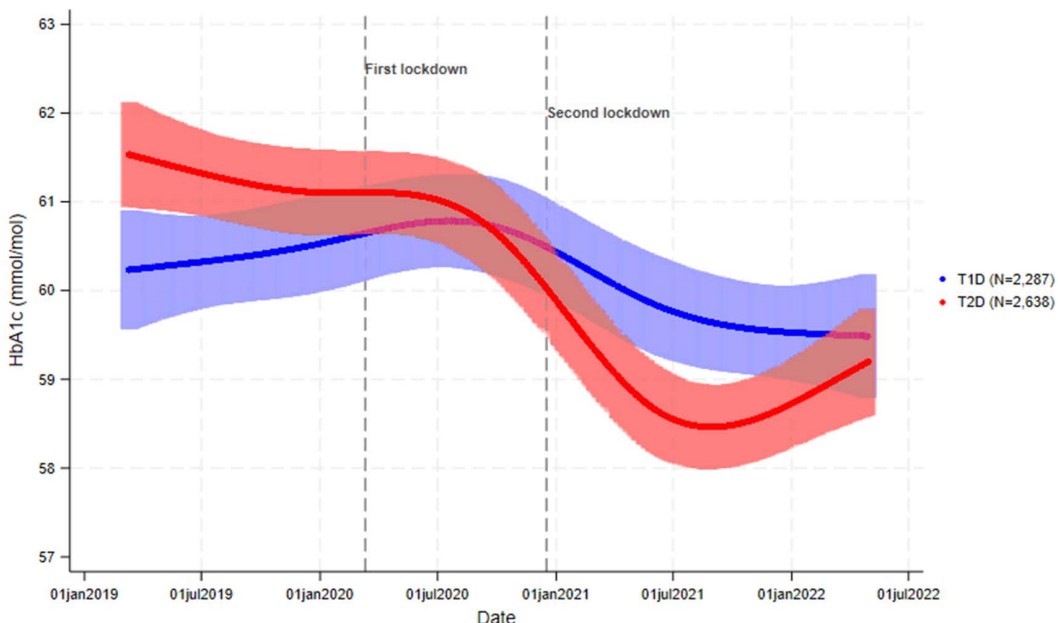

**Fig 2. Trajectories of HbA1c before and during COVID-19 according to diabetes type (type 1-diabetes and type 2-diabetes).**

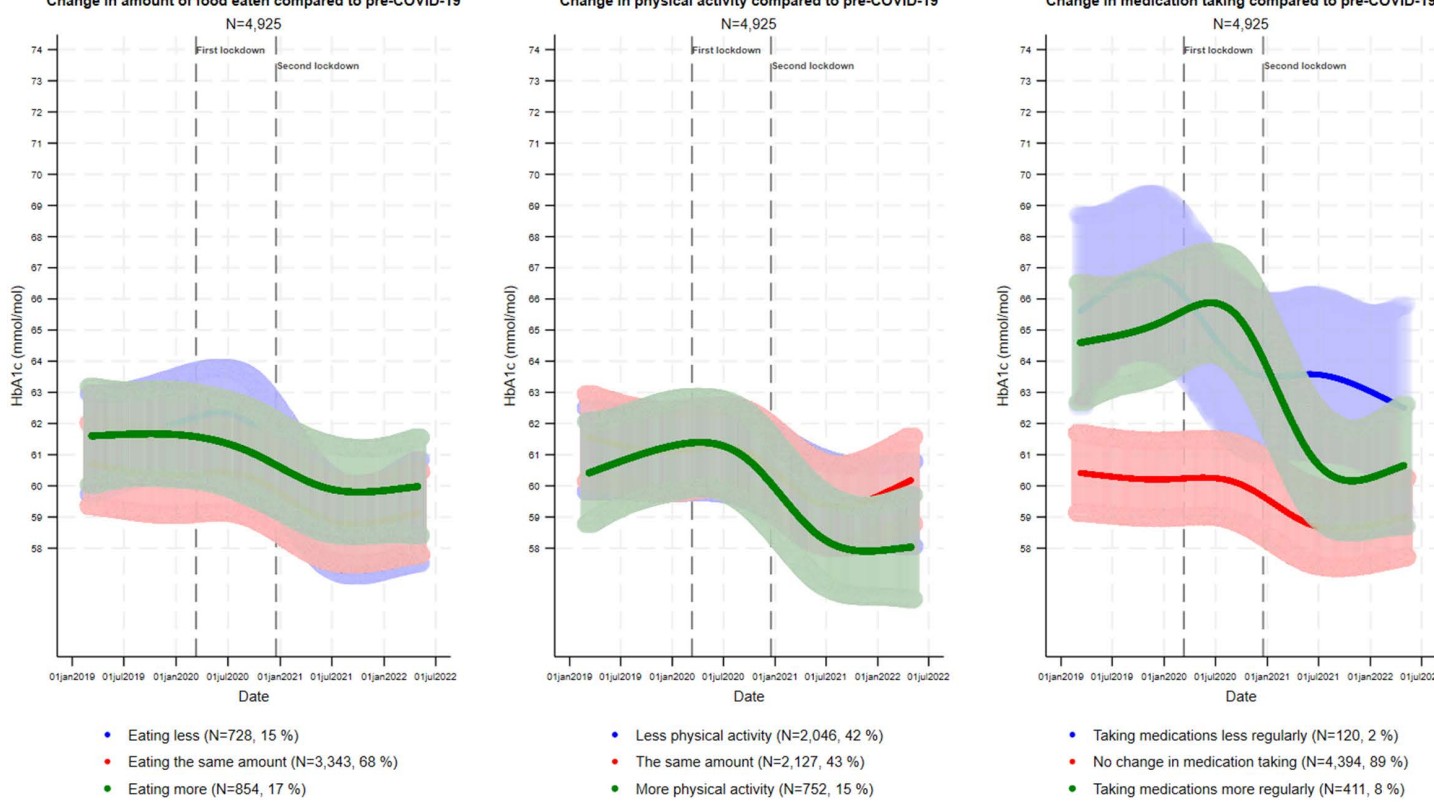

**Fig 3. Self-reported change in diet, physical activity and medication taking compared to pre-COVID-19 in relation to trajectories of HbA1c in people with diabetes (N = 4,925).** Adjusted for age (yrs), sex (male/female), diabetes duration (yrs), number of diabetes-related complications (0, 1, 2, ≥ 3) and educational attainment (elementary, high school or vocational, short or bachelor level, master's degree or higher, other/not declared). The curves are estimated for a 50-year-old female who has had diabetes for 15 years, who has two diabetes-related complications and a short or bachelor level educational attainment. The frequencies within each category of the change variables are determined based on the data in the first imputed dataset as an example.

We observed statistical interactions between HbA1c trajectories and categories of each exposure variable (all p < 0.001), although the trajectories appeared to differ only between the categories related to medication taking. Those reporting more regular medication taking showed a decrease in HbA1c from approximately 65/66 (8.1/8.2%) mmol/mol to approximately 60/61 mmol/mol (7.6/7.7%) from the summer of 2020 and throughout follow-up. The extent of the decrease in HbA1c in those reporting more regular medication taking was overall similar in people with T1D and people with T2D, although people with T2D appeared to have a larger decrease followed by a slight increase from July of 2021 throughout follow-up (S2 Appendix). Those who reported *less* regular medication taking also exhibited a decrease in HbA1c, although not to the same extent as those reporting more regular medication taking. Notably, both the group reporting less and the group reporting more regular medication taking had the highest mean HbA1c pre-COVID-19. Confidence intervals were generally wide. The sensitivity analyses supported the findings in the main analyses (S2 Appendix).

## Discussion

In almost 5,000 adults visiting diabetes outpatient clinics, we found that during COVID-19 HbA1c decreased in adults with T2D, while no change was observed in adults with T1D during COVID-19. During the first year of COVID-19 the majority upheld most of their diabetes self-management routines, although many decreased physical activity levels. For both

people with T1D and T2D we found different HbA1c trajectories according to self-reported change in diet, physical activity and medication taking. We observed a noteworthy decrease in HbA1c of approximately 5 mmol/mol (2.6%) (from 65/66 mmol/mol (8.1/8.2%) during the summer of 2020) during COVID-19 both in people with T1D and people with T2D, in those reporting more regular medication taking.

Our findings suggest that during COVID-19, HbA1c levels decreased slightly in people with T2D and in those with other diabetes types, while they appeared unchanged in people with T1D. According to systematic reviews of studies from multiple countries, the evidence is not clear regarding the direction of change in HbA1c levels during the pandemic in people with diabetes; it may have increased in people with T2D [20,21], which is inconsistent with our findings, but decreased in people with T1D [20]. Other studies suggest no change in HbA1c [22]. Although natural disasters, i.e., other external events than the COVID-19 pandemic, have been associated with higher glycemic levels in people with T1D [27,28], the impact is likely to be highly individual and to some extent dependent on changes in diabetes self-management. In a Danish study, almost one out of four people with diabetes reported that during COVID-19 their diabetes had become harder to manage, while others reported that their ability to self-manage their diabetes was unchanged [16]. Therefore, the pandemic posed a challenge for a non-negligible number of people with diabetes in Denmark, which may have impacted their glycemic levels.

Consistent with other studies [7–15] we found that a noteworthy number of people with diabetes reported less physical activity during COVID-19. This is consistent with a review of studies of the general population, showing that physical activity decreased and sedentary behavior increased during COVID-19 [29]. Being quarantined and taking safety precautions to avoid illness may have prompted a more physically inactive lifestyle, with increased screen time at home and less leisure activities, e.g., less time spent in fitness centers [29]. Moreover, vast structural societal changes during the first year of COVID-19 influenced the everyday life of people with diabetes and society at large, including closing of non-essential services, e.g., restaurants and food markets in some countries [6], which may also have impacted people's diet. We found that one in ten with T1D and one in five with T2D reported decreased food intake. Also, we found that one in five with T1D and approximately one in seven with T2D reported increased food intake. A systematic review of studies, primarily from Asian countries, found that people with diabetes reported both positive (increased fruit and vegetable and decreased animal protein and alcohol consumption) and negative (increased consumption of snacks and sweets) dietary changes [6]. The COVID-pandemic may for some have been a window of opportunity for improved diabetes self-management, while for others the change in circumstances made diabetes self-management more difficult.

Our findings suggest that during COVID-19, HbA1c trajectories differed according to changes in diet intake and physical activity in people with diabetes. Nevertheless, the trajectories for these two exposure variables did to some extent align with the overall trend for each diabetes type. Thus, despite less physical activity and higher food intake during the pandemic, these changes did not appear to translate into changes in glycemic levels. According to our assessment of the construct validity of the variables change in diet and change in physical activity they appeared valid. However, if we had included more accurate and comprehensive dietary assessments and objective assessments of physical activity, e.g., using TriAxial Accelerometers [30], the groupings and thus trajectories might have turned out differently. We found that almost 200 people with T1D (8.5%) and 224 people with T2D (8.5%) reported more regular medication taking, which was associated with a clinically relevant HbA1c decrease. The decrease in HbA1c started approximately three months after the national lockdown in March 2020 and shortly after diabetes was defined as a high-risk disease by the Danish Health Authority (7th of April) [31]. Given that HbA1c reflects average blood sugar levels 2–3 months prior, this decrease may be reflective of changes to medication following the major events early in the COVID-19 pandemic. Being quarantined and having fewer social obligations may have motivated some to increase medication taking relative to pre-COVID-19. Improved medication taking, together with improved glycemic levels, during COVID-19 has also been reported in other samples of people with diabetes [19]. However, we found that HbA1c appeared to increase after July 2021 in people with T2D, which may reflect a return to pre-pandemic diabetes self-management behaviors. Perhaps surprisingly, people with

T2D who reported less regular medication taking also had an improvement in HbA1c at the end of follow-up. Those who reported less regular medication taking and those who reported more regular medication taking had a similar mean HbA1c pre-pandemic, which was higher than the pre-pandemic HbA1c for those who reported no changes in medication taking. It is difficult to pinpoint why those reporting less regular medication taking appeared to improve their HbA1c. However, those reporting less regular medication taking constituted the smallest group, and the finding that they reduced their HbA1c may to some extent be spurious. The uncertainty associated with the estimates of HbA1c for those reporting less regular medication taking is reflected in the very wide confidence intervals for this group. Alternative explanations could be mis-classification of the exposure or regression towards the mean. Overall, these findings suggest that improved medication taking rather than improved diet and physical activity has the potential to positively impact glycemic levels in this primarily middle-aged and elderly population of people with diabetes.

### Limitations and strengths

This study has some limitations. Self-reporting of diabetes self-management is likely prone to information bias, i.e., some may find it difficult to recall and compare their self-management behaviors before and during the pandemic. Also, residual, e.g., from lack of data on diabetes technology and medication regime, and unknown confounding cannot be ruled out. How-ever, reports from clinics in our own department (Steno Diabetes Center Odense, Odense University Hospital) indicate that very few patients changed their medication regimen during the COVID-19 lockdown. Changes were primarily made during acute illness. Moreover, in this study, we conducted sensitivity analyses, where we restricted to patients with regular HbA1c assessment, which we suggest reflects the patient population with the highest treatment intensity. The sensitivity analyses generally confirmed our main results. Therefore, confounding due to changes in treatment regimens (such as medication adjustments or the introduction of diabetes technology) is not expected to be an issue. Moreover, while not a limitation per se, our sample included only people attending diabetes outpatient clinics and our findings may thus not be generalizable to all people with diabetes. Strengths of this study are the linkage between self-reported changes in self-management behav-iors during the pandemic and clinical data, the relatively large sample size and that we could separate our analyses accord-ing to diabetes type. Moreover, a large proportion of our sample had HbA1c data available for analyses.

### Conclusions

HbA1c trajectories during and after COVID-19 differed between diabetes types. In general, HbA1c trajectories varied according to self-reported changes during the first year of COVID-19 in diet, physical activity and medication taking in both diabetes types, but glycemic levels appeared to be improved mostly in those who reported more regular medication intake. Our findings suggest that the pandemic impacted some components of diabetes self-management, which to some extent translated into improved glycemic levels. This is valuable information for clinicians in diabetes care about how structural changes impact the lives of people with diabetes and about relevant areas for intervention to decrease HbA1c levels. Future pandemic preparedness should prioritize structural adaptations and policy frameworks to ensure uninter-rupted and equitable diabetes care.

### Supporting information

**S1 Appendix. Questionnaire.** The questionnaire used to collect data on diabetes self-management and personal char-acteristics. The questionnaire is in Danish.
(DOCX)

**S2 Appendix. Supplementary analyses.** Supplementary analyses include descriptive statistics and analyses according to diabetes type.
(DOCX)

## Acknowledgments

We would like to thank statistician Sören Müller from the organization Open Patient data Explorative Network (OPEN) Registry and Statistics for his valuable statistical support. Moreover, we would like to thank OPEN for general research support.

## Author contributions

**Conceptualization:** Martin Gillies Banke Rasmussen, Emilie Just-Østergaard, Jacob Volmer Stidsen, Ingrid Willaing, Grete Skøtt Pedersen.

**Data curation:** Martin Gillies Banke Rasmussen, Emilie Just-Østergaard, Ingrid Willaing, Grete Skøtt Pedersen.

**Formal analysis:** Martin Gillies Banke Rasmussen.

**Funding acquisition:** Martin Gillies Banke Rasmussen.

**Investigation:** Martin Gillies Banke Rasmussen, Emilie Just-Østergaard, Jacob Volmer Stidsen, Ingrid Willaing, Grete Skøtt Pedersen.

**Methodology:** Martin Gillies Banke Rasmussen, Emilie Just-Østergaard, Jacob Volmer Stidsen, Ingrid Willaing, Grete Skøtt Pedersen.

**Project administration:** Martin Gillies Banke Rasmussen.

**Software:** Martin Gillies Banke Rasmussen.

**Supervision:** Grete Skøtt Pedersen.

**Validation:** Martin Gillies Banke Rasmussen.

**Visualization:** Martin Gillies Banke Rasmussen.

**Writing – original draft:** Martin Gillies Banke Rasmussen.

**Writing – review & editing:** Martin Gillies Banke Rasmussen, Emilie Just-Østergaard, Jacob Volmer Stidsen, Ingrid Willaing, Grete Skøtt Pedersen.

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
