## [Decision Letter · Decision Letter 0]

28 Nov 2024

PONE-D-24-33504What is the impact of structural changes in society on diabetes self-management and trajectories of HbA1c? A cohort study before, during and after the COVID-19 pandemic in people with diabetes treated at outpatient clinicsPLOS ONE?

Dear Dr. Rasmussen,

We look forward to receiving your revised manuscript.

Kind regards,

Andrea Da Porto

Academic Editor

PLOS ONE

Journal Requirements:

Reviewers' comments:

Reviewer's Responses to Questions

**Comments to the Author**

1. Is the manuscript technically sound, and do the data support the conclusions?

Reviewer #1: Yes

Reviewer #2: Partly

2. Has the statistical analysis been performed appropriately and rigorously?

Reviewer #1: Yes

Reviewer #2: I Don't Know

3. Have the authors made all data underlying the findings in their manuscript fully available?

Reviewer #1: No

Reviewer #2: No

4. Is the manuscript presented in an intelligible fashion and written in standard English?

Reviewer #1: Yes

Reviewer #2: Yes

Reviewer #1: Thank you for your manuscript.

Manuscript can be strengthened further by adding a brief description of COVID19 impact in your country, what levels of lockdowns were implemented, that is whether community dwelling individuals were allowed to exercise outdoors?, what happened to regular outpatient services, whether there was still physical face to face clinics or telehealth?

Was there any shortage of medications during COVID19?

Proportion of HbA1c available appears to be very high which is the strength of the data and manuscript. do we have any comparison data from other countries?

In the final conclusion, could you please add a brief paragraph on structural changes to diabetes care during future pandemics, how policy should be in place to deal with chronic conditions.

Reviewer #2: Thank you for allowing me to review this submission focusing on results of a survey (Denmark) regarding diabetes self-management and A1C prior to, during and after the COVID-19 pandemic.

Overall, the information presented is organized and relatively straightforward. The manuscript would benefit from editing for English grammar.

Abstract

L33 – could the authors please rewrite this sentence to clarify changes in physical activity: Suggest something like – “No changes in physical activity were reported by 4X% while 4Y% reported decreased physical activity.”

L35 – suggest “taking medication more regularly”

Statistical analyses: I am not a statistician and recommend a statistical review if available.

The finding that improved compliance with medication (taking medication more regularly) rather than activity and diet was a key to success may be an appropriate conclusion that may be helpful for patients and healthcare professionals. However, the major data challenge here is that A1C also improved for those taking medication less regularly! The explanation (L279-284) that “perhaps medication taking during [pandemic]…improved for both groups” doesn’t fit with the results of the survey (i.e. when these participants reported taking medication less regularly). This makes me wonder about a seasonal affect (summer increase in activity when able to be outside in 2021?) or other confounders/factors below.

My major remaining questions regarding the results relate to the following (could the authors please comment, I realize they may not have access to this information, but it would have been very helpful – they mention “technology” briefly, but a lot has changed in the past few years:

1) Whether the survey asked about changes in weight (kg) or whether this data is available – it seems that weight might be a way to validate changes in diet/activity.

2) What medications were patients treated with overall? Are those data available?

a. In particular, what was the availability of GLP-1 medications in Denmark during this time (had they become widely available to patients with T2DM? – this could be a major confounder or important explanation).

3) What was availability of personal Continuous Glucose Monitoring Systems (CGM) during the pandemic in Denmark? – This technology has become increasingly available over the last few years and is generally associated with improvements in A1C (~1%).

4) Was telemedicine (televisits) available to patients?

Table 1. Please explain the PAID-5 and WHO-5 scores in the methods/results for readers who might not be familiar with the details of those scores.

**Do you want your identity to be public for this peer review?** For information about this choice, including consent withdrawal, please see our Privacy Policy

Reviewer #1: **Yes: ** A/prof Shamasunder Acharya

Reviewer #2: No

---

## [Author Response · Author response to Decision Letter 1]

22 Jan 2025

Dear reviewers

Thank you very much for taking the time to review our manuscript submitted to PLOS ONE. We address each of your comments below and believe that the changes made in response to your feedback have significantly improved the quality of the manuscript. We are sincerely grateful for your valuable input.

Reviewer #1:

“Thank you for your manuscript.”

Thank you for the specific comments below. In particular, thank you for pointing out the need for providing more context for the study, as well as pointing out the need to highlight in the conclusion one of the important lessons of the study.

“Manuscript can be strengthened further by adding a brief description of COVID19 impact in your country, what levels of lockdowns were implemented, that is whether community dwelling individuals were allowed to exercise outdoors?, what happened to regular outpatient services, whether there was still physical face to face clinics or telehealth?

Was there any shortage of medications during COVID19?”

We acknowledge that it is necessary to provide some context for the study. Thus, we have added some sentences in the introduction where we explain some of the major structural changes in Denmark during COVID-19, including the change from face-to-face to telehealth contacts in Danish hospital setting.

As far as we know, there were no specific shortages of medication in Denmark during COVID-19.

“Proportion of HbA1c available appears to be very high which is the strength of the data and manuscript. do we have any comparison data from other countries?”

Thank you for highlighting this strength of the study, which we have specified in the discussion.

As far as we know, there is no review which has assessed the variation in HbA1c testing and thus availability of HbA1c data for research across countries. However, this is most likely going to vary from country to country.

“In the final conclusion, could you please add a brief paragraph on structural changes to diabetes care during future pandemics, how policy should be in place to deal with chronic conditions.”

Thank you for this suggestion. This is an important lesson from this study. A new final sentence has been added.

Reviewer #2:

“Thank you for allowing me to review this submission focusing on results of a survey (Denmark) regarding diabetes self-management and A1C prior to, during and after the COVID-19 pandemic.

Overall, the information presented is organized and relatively straightforward. The manuscript would benefit from editing for English grammar.”

Thank you very much for providing specific suggestions for changes in sentences as well as asking questions based on an in-depth study of the statistical analyses presented in the manuscript. We have adjusted the manuscript where possible.

Although all authors have thoroughly read the manuscript prior to submission, for the re-submission the manuscript has been edited by a native English speaking person.

“Abstract

L33 – could the authors please rewrite this sentence to clarify changes in physical activity: Suggest something like – “No changes in physical activity were reported by 4X% while 4Y% reported decreased physical activity.””

We discussed this sentence at several occasions. Thank you for the specific suggestion. We have changed the sentence to match your suggestion.

“L35 – suggest “taking medication more regularly””

We have also changed this sentence according to your suggestion.

“Statistical analyses: I am not a statistician and recommend a statistical review if available.”

The analyses were conducted with guidance from a statistician employed at a research support unit in our geographical region (the Region of Southern Denmark). We have acknowledged the statisticians’ support in the Acknowledgement section.

“The finding that improved compliance with medication (taking medication more regularly) rather than activity and diet was a key to success may be an appropriate conclusion that may be helpful for patients and healthcare professionals. However, the major data challenge here is that A1C also improved for those taking medication less regularly! The explanation (L279-284) that “perhaps medication taking during [pandemic]…improved for both groups” doesn’t fit with the results of the survey (i.e. when these participants reported taking medication less regularly). This makes me wonder about a seasonal affect (summer increase in activity when able to be outside in 2021?) or other confounders/factors below.”

We find it difficult to account for this finding. However, we acknowledge that it should be elaborated in the manuscript. We also changed the sentence you refer to. The necessary changes have been made to the discussion. We find that the main concern with this result is that the group in question (those who report less regular medication taking) is the smallest group (N=120) among the three medication groups and the estimates for this group thus may suffer from low statistical power. This is reflected in confidence intervals being widest for this group. Therefore, the finding may to some extent be spurious.

Another potential explanations may be misclassification of the exposure, regression towards the mean or perhaps residual or unknown confounding. We have added these potential explanations to the discussion.

“My major remaining questions regarding the results relate to the following (could the authors please comment, I realize they may not have access to this information, but it would have been very helpful – they mention “technology” briefly, but a lot has changed in the past few years:”

We thank for you for these in-depth comments. Unfortunately, several of the variables that you inquire about were not available for this study. In the limitations section we acknowledge that residual or unknown confounding from not having these variables may be an issue.

However, reports from our own clinic (Steno Diabetes Center Odense, Odense University Hospital) indicate that very few patients changed their medication regimen during the COVID-19 lockdown. Changes were primarily made during acute illness. Therefore, confounding due to changes in treatment regimens (such as medication adjustments or the introduction of diabetes technology) is not expected to be an issue. This is now elaborated in the discussion.

“1) Whether the survey asked about changes in weight (kg) or whether this data is available – it seems that weight might be a way to validate changes in diet/activity.”

Unfortunately, we did not collect data specifically on changes in kg bodyweight, although this would have been an important piece of information to validate against what was reported for diet and physical activity. Nor can we ascertain this data from registries. However, in the questionnaire we included a question about whether the respondents’ bodyweight changed with no specification of the magnitude of change in kg (the questionnaire is included in Appendix 1, although, unfortunately, it is in Danish). Below we have tabulated this variable against the variables ‘Changes in diet’ and ‘Changes in physical activity’:

Eating less Eating the same amount Eating more

Increased bodyweight, n (%) 103 (13.6) 780 (22.2) 602 (66.7)

No change in bodyweight, n (%) 201 (26.5) 2,164 (61.6) 202 (22.4)

Decreased bodyweight, n (%) 455 (60.0) 570 (16.2) 99 (11.0)

Less physical activity The same amount More physical activity

Increased bodyweight, n (%) 961 (44.6) 366 (16.4) 158 (20.0)

No change in bodyweight, n (%) 775 (36.0) 1,454 (65.2) 338 (42.7)

Decreased bodyweight, n (%) 417 (19.4) 411 (18.4) 296 (37.4)

It would be better (i.e. more precise) to compare the two variables to actual changes in bodyweight and thus the above should be interpreted with some caution. With that in mind the distribution for change in amount of food eaten generally reflect what we would expect if change in food eaten was assessed correctly, e.g. among those “eating less” almost two-thirds reported a “Decreased bodyweight”. The same appears to be the case for changes in physical activity, although the distributions are less “pronounced” (although it may be difficult to determine the “correct” distributions). However, the “direction” of the distributions appear to be what one would expect if change in physical activity was assessed correctly, e.g. among those reporting “Less physical activity” almost 45 % report “Increased bodyweight”.

We hope that the above sheds some light on the validity of our assessments of changes in food intake and changes in physical activity in the questionnaire.

We have elaborated on these comparisons in the manuscript (methods, results and discussion). The tables themselves have been included in 2nd supplementary material.

“2) What medications were patients treated with overall? Are those data available?

a. In particular, what was the availability of GLP-1 medications in Denmark during this time (had they become widely available to patients with T2DM? – this could be a major confounder or important explanation).”

Unfortunately, we did not have these data available for the study. However, we tried to mimic treatment intensity by conducting analyses specifically for those with more regular HbA1c assessments, i.e. we restricted our analyses to the subsample of patients who had their HbA1c assessed on a regular basis during COVID-19. These patients thus had regular consultations and we would expect changes to the treatment regimen only in this sample. The sensitivity analyses generally confirmed our main results, i.e. “treatment intensity” does not appear to confound our main findings. We have introduced these considerations to the discussion of the paper.

“3) What was availability of personal Continuous Glucose Monitoring Systems (CGM) during the pandemic in Denmark? – This technology has become increasingly available over the last few years and is generally associated with improvements in A1C (~1%).”

We would like to kindly refer to the previous answer for medication taking.

”4) Was telemedicine (televisits) available to patients?”

Yes, the Danish healthcare system made a radical shift from face-to-face consultations to telehealth consultations. Reviewer #1 requested more context to the study. Therefore, we have now mentioned this shift in the introduction.

“Table 1. Please explain the PAID-5 and WHO-5 scores in the methods/results for readers who might not be familiar with the details of those scores.”

We have added text in the methods section explaining these scales for those unfamiliar with them. Thank for noting that these descriptions were lacking in the methods section.

---

## [Decision Letter · Decision Letter 1]

16 Jul 2025

What is the impact of structural changes in society on diabetes self-management and trajectories of HbA1c? A cohort study before, during and after the COVID-19 pandemic in people with diabetes treated at outpatient clinics

PONE-D-24-33504R1

Dear Dr. Rasmussen,

We’re pleased to inform you that your manuscript has been judged scientifically suitable for publication and will be formally accepted for publication once it meets all outstanding technical requirements.

Kind regards,

Wen-Jun Tu

Academic Editor

PLOS ONE

Additional Editor Comments (optional):

Reviewers' comments:

Reviewer's Responses to Questions

**Comments to the Author**

Reviewer #2: All comments have been addressed

2. Is the manuscript technically sound, and do the data support the conclusions?

Reviewer #2: Partly

3. Has the statistical analysis been performed appropriately and rigorously?

Reviewer #2: I Don't Know

4. Have the authors made all data underlying the findings in their manuscript fully available?

Reviewer #2: No

5. Is the manuscript presented in an intelligible fashion and written in standard English?

Reviewer #2: Yes

Reviewer #2: The manuscript is much improved and conclusions and limitations are more clearly spelled out. Thank you to the authors for addressing concerns in detail. I think the authors' conclusion that vigilance may have been an important factor in medication adherence makes sense. The psychology of adherence is complex, but heightened concern early in the pandemic followed by relaxation as the pandemic waned would fit with fluctuating adherence in patients. Following on from that, I wonder whether patients who reported being vaccinated (a surprisingly low percentage) were much more likely to have improved A1Cs.

**Do you want your identity to be public for this peer review?** For information about this choice, including consent withdrawal, please see our Privacy Policy

Reviewer #2: No

---

## [Editor Report · Acceptance letter]

PONE-D-24-33504R1

PLOS ONE

Dear Dr. Rasmussen,

I'm pleased to inform you that your manuscript has been deemed suitable for publication in PLOS ONE. Congratulations! Your manuscript is now being handed over to our production team.

Kind regards,

on behalf of

Dr. Wen-Jun Tu

Academic Editor

PLOS ONE